# A Comparative Analysis of Gender Discrepancy Stress, Attitudes toward Intimate Partner Violence, and Perpetration among Young Adults in the USA and Uganda

**DOI:** 10.3390/ijerph192013373

**Published:** 2022-10-17

**Authors:** Matthew J. Lyons, Monica H. Swahn, Rachel Culbreth, Dennis Reidy, Tina Musuya, Paul Bukuluki

**Affiliations:** 1Wellstar College of Health and Human Services, Kennesaw State University, Kennesaw, GA 30144, USA; 2American College of Medical Toxicology, Phoenix, AZ 85028, USA; 3School of Public Health, Georgia State University, Atlanta, GA 30302, USA; 4Social Development Direct, Kampala 759125, Uganda; 5Department of Social Work and Social Administration, Makerere University, Kampala 759125, Uganda

**Keywords:** intimate partner violence, gender discrepancy stress, violence, alcohol, Uganda

## Abstract

Background: There is a dearth of data on the modifiable factors that contribute to violence in low- and middle-income countries, including attitudes regarding intimate partner violence (IPV) and perceptions of gender identity. We examined these factors using a cross-cultural comparison between young adults in Uganda and the United States. Methods: A cross-sectional survey was distributed to young adults aged 18 to 25 in Uganda (n = 300) and the U.S. (n = 300). Survey questions assessed demographics, attitudes toward IPV, IPV victimization and perpetration, gender discrepancy, discrepancy stress, and alcohol use. We conducted chi-square tests, as well as bivariable and multivariable logistic regression analyses, separately for participants in each country. Results: The prevalence of IPV perpetration differed significantly by country for men (58.06% in the U.S. vs. 42.73% in Uganda; *p* = 0.03) and women (40.00% in the U.S. vs. 14.00% in Uganda; *p* < 0.01). IPV victimization differed by country for men (67.74% in the U.S. vs. 51.82% in Uganda; *p* = 0.02) but not for women. Gender discrepancy and discrepancy stress also varied by country and by sex and were higher in the U.S. for both men and women. IPV victimization was a common risk factor for adults in both Uganda (Adj. OR = 23.47; 95% CI: 7.79, 70.22) and the U.S. (Adj. OR = 27.40; 95% CI: 9.97, 75.32). In Uganda, male sex was significantly associated with IPV perpetration in multivariable analyses (Adj. OR = 6.23; 95% CI: 2.45, 15.86), and so were IPV attitudes (Adj. OR = 2.22; 1.20, 4.10). In the U.S., a likely alcohol use disorder (AUD) was also significantly associated with IPV perpetration (Adj. OR = 7.11; 95% CI: 2.25, 22.54). Conclusions: Permissive IPV attitudes were associated with IPV perpetration among Ugandan participants, while likely AUD was associated with perpetration in U.S. participants. Overall, IPV perpetration was significantly higher for U.S. males compared with Ugandan males. These findings indicate that cultural adaptations to global IPV interventions may be necessary to respond to differing needs in different countries.

## 1. Introduction

Intimate partner violence (IPV) has significant detrimental consequences for both health and human rights, and it is highly prevalent both in the United States and across Sub-Saharan Africa (SSA) [1,2,3,4,5]. In the U.S., more than 43.6 million, or 36.4% of women, reported having experienced physical violence, sexual violence, stalking, or some combination of the three perpetrated by their intimate partner at some point in their lives [5]. In SSA, data indicate that the prevalence of violence is similarly high [4], with tremendous burden to health and well-being, particularly among vulnerable women [6,7]. This is also the case in Uganda, where IPV and gender-based violence are common among vulnerable youth who reside in urban slums [8] and where violence is often exacerbated by the link to both alcohol use and human immunodeficiency virus (HIV) [9].

A seminal community-based intervention known as SASA! was implemented and evaluated in Kampala, Uganda [10,11]. This ground-breaking study sought to address gender-based violence and HIV through community mobilization, which entails developing a group of community activists, encouraging those activists to interrogate imbalances of power in relations between men and women, and fostering the power of those activists to create change within their own community. This process, which entails four strategies (activism, media advocacy, communication materials, and training), was found to be effective in a cluster randomized controlled trial [12]. The key metric used in the study was the social acceptability of IPV, a concept that is rarely studied or compared across cultural contexts in low-resource settings, but which is clearly relevant in the development and implementation of violence prevention programs regardless of setting and context.

Gender norms have been linked to the prevalence of IPV, with adherence to traditional notions of masculinity among male partners predicting IPV perpetration and non-adherence to traditional feminine roles among female partners predicting higher risk of IPV perpetration among hypermasculine men [13]. In Uganda, one recent study analyzing cultural gender norms indicated a high prevalence of gender stereotyping and significant power imbalances between adolescent girls and boys [14]. In turn, recent work has shown that gender norms in Uganda are related to adolescent sexual behavior, with gender norms being associated with sexual behaviors (with more equitable norms associated with safer behaviors) [15] and with family planning behaviors [16]. While these cultural contextual factors are likely implicated in the complex causal chain that leads to IPV, very little comparative work has been done to explore how those factors and their impact on IPV may differ internationally.

Gender discrepancy stress, or a feeling of unease associated with self-perceived non-adherence to socially constructed gender roles, has also been implicated as a potential risk factor for IPV perpetration, particularly among men in the U.S. [17,18,19,20]. A recent systematic review of cross-cultural differences in IPV attitudes among tertiary students indicated that U.S. students are less likely to have permissive attitudes toward IPV than those of students in other countries [21]. However, cross-country studies remain relatively scarce, particularly comparing populations in high-income versus low- and middle-income countries. In Uganda, perceptions among men that males dominate decision making over household spending have been shown to be associated with higher self-reported IPV perpetration, and male perceptions that such decisions are shared or made independently by women have been shown to be associated with lower probability of perpetration [22]. Moreover, research in Uganda has shown a high prevalence of permissive attitudes toward IPV and an association between having witnessed IPV perpetration and permissive attitudes [23].

While attitudes toward IPV and gender, as well as experiences of gender discrepancy and discrepancy stress, are implicated in the etiology of IPV, perpetration occurs within a broader social-ecological and economic context. An individual instance of IPV perpetration is likely to occur in response to a complex set of contextual and individual factors, including but not limited to IPV attitudes and gender discrepancy. IPV perpetration, therefore, cannot be fully understood without reference to a range of other co-present factors. For example, socioeconomic status (SES) has been shown to be associated with IPV perpetration and risk across settings, with one study across 10 countries showing a protective effect of high SES against IPV victimization [24]. Similarly, a large study of women across low-income countries demonstrated a negative association between household wealth and IPV prevalence [25]. There is also a clear association between alcohol consumption and IPV that has been examined for both male-to-female perpetration and female-to-male perpetration. These associations appear to be stronger in studies that include measurement of heavy alcohol consumption [26]. In the U.S., one seminal study found that among men who perpetrated IPV, between 30% and 40% reported being under the influence of alcohol at the time of perpetration [27]. In Uganda, one study demonstrated that alcohol use was associated with increased incidence of both physical violence and sexual coercion [28]. Another study of youth drinkers in Uganda found that 51% of boys and 41% of girls reported that they got into a fight as a result of alcohol use [29].

Despite significant research and intervention efforts, relatively few studies provide comparative context for how the correlates of IPV may differ across countries and cultures [30,31], particularly among young adults, who are at elevated risk for both IPV perpetration and victimization [32,33]. Within this limited body of literature, cross-cultural comparisons including SSA have been even more scarce [34]. Further, gender discrepancy stress and norms supporting IPV are important factors that may differ between the U.S. and SSA. As the SASA! study demonstrated in urban Kampala, a community-based intervention can shift the norms that support violence (and HIV transmission) [35,36,37], and these findings may have important implications for other settings, including the U.S.

To respond to the dearth of international comparisons with respect to the association between IPV attitudes, gender discrepancy stress, IPV victimization, and IPV perpetration, we conducted this study to explore these associations among young adults in the U.S. and Uganda. We selected these two countries in order to differentiate IPV attitudes between a high-income country (U.S.) and a low-income country (Uganda) and to determine specific needs for tailoring interventions. IPV remains a priority topic for violence prevention in the U.S., and there has been significant investment and research conducted on IPV victimization and perpetration. The opposite is the case in Uganda, where there are relatively few studies on violence prevention and potential mitigating factors (aside from the SASA! study). As such, if the patterns observed with respect to IPV attitudes, gender-discrepancy stress, IPV victimization, and IPV perpetration are similar across the two countries, some of the existing research already conducted in the U.S. may be extended more easily and adapted for the Ugandan cultural context.

The research questions that guided this cross-sectional survey research study were as follows: (1) is the prevalence of IPV perpetration and victimization similar among young adults in the U.S. and in Uganda?; (2) are gender differences in IPV perpetration and victimization similar in the U.S. and in Uganda?; (3) are IPV attitudes more strongly associated with IPV perpetration in Uganda than in the U.S.?; and (4) is gender discrepancy stress more strongly associated with IPV perpetration in Uganda than in the U.S.? These comparisons can inform the development of more targeted interventions and help to determine if intervention strategies may be applicable across settings. The purpose of this paper, therefore, is to present a comparative analysis of the associations between gender discrepancy stress, attitudes toward intimate partner violence, and intimate partner violence perpetration between young adults in the U.S. and Uganda.

## 2. Materials and Methods

Qualtrics distributed the brief online survey to young adults, aged 18 to 25 years old, during the spring of 2021. All participants provided consent at the beginning of the survey. The only two eligibility criteria pertained to country of residence (Uganda or the U.S.) and age (18–25). This cross-sectional online survey was distributed in English, which is the official language in Uganda. The study population consisted of 600 participants in both Uganda (n = 300) and the U.S. (n = 300). Overall, 47.9% of participants were female. The Qualtrics Online Survey Panel team handled participant recruitment/sampling and distribution of the survey. Given the third-party recruitment strategy, no response rate can be computed. Those who participated in the survey received compensation in an amount based on Qualtrics’ discretion and agreed upon by participants prior to taking the survey. The study was approved by the Georgia State University IRB.

The survey contained 8 modules with a total of 95 questions and took about 20 min to complete. The eight modules were: Consent/Demographics, Marketing Influence, Uganda Alcohol Ad Exposure, Alcohol Use Assessment, Gender Roles, Attitudes Towards Physical and Sexual Dating Violence, General Violence Measure, and the Brief Resilience Scale.

### 2.1. Sample

The analytic sample excluded participants who had never been in a relationship (Uganda n = 89; United States n = 99). A series of standard questions was included in the survey to determine basic demographic characteristics in the sample and to collect important control variables. Among these were participant sex, education level, self-rated health, and perceived social status as a proxy for SES. In the initial sample, 5 Ugandans and 8 Americans gave responses other than “male” or “female” for their gender. Of those who responded other than “male” or “female”, 4 Ugandans and 5 Americans had also never been in a relationship. Given the small number of such participants remaining, the analytic sample included only those participants who responded either “male” or “female”. The analytic samples therefore included 210 participants from Uganda and 198 participants from the U.S.

### 2.2. Measures

#### 2.2.1. Social Status

An adaptation of the Macarthur Scale of Subjective Social Status–Adult Version was used to assess self-reported social status as a proxy for SES [38]. For this measure, participants were presented with an image of a ruler with markings from 0 to 100 and asked to imagine that the left side of the ruler (0) represented those in society with the least money, least education, and least respected jobs, while the right side (100) represented those in society with the most money, most education, and most respected jobs. Participants were then instructed to select the location on the ruler which they felt best represented them, generating perceived social status scores ranging from 0 to 100, with 50 representing the midpoint.

#### 2.2.2. Attitudes toward Dating Violence

Attitudes toward IPV were assessed using items from the Attitudes Toward Dating Violence Scales [39]. To minimize the burden and time requirements of the surveys, only select items from the Attitudes Toward Dating Violence Scales were included. Among male participants a selection of ten items from the physical and sexual violence subscales were used, and among female participants a selection of six items from the same two scales were used. Each question began by saying, “Please share your thoughts and feelings on how couples may interact”. For men, the ten items that followed were: “Some partners deserve to be slapped by their boyfriends”; “Sometimes guys just cannot stop themselves form punching partners”; “When a guy pays on a date, it is O.K. for him to pressure his partner for sex”; “Sometimes a guy cannot help hitting his partner when they make him angry”; “When guys get really sexually excited, they cannot stop themselves from having sex”; “Sometimes jealousy makes a guy so crazy that he must slap his partner”; “Guys should never get their partners drunk to get them to have sex”; “Sometimes love makes a guy so crazy that he hits his partner”; “A guy should not touch his partner unless they wanted to be touched”; and “It is O.K. for a guy to slap his partner if they deserve it”. For women, the items that followed were: “It is O.K. for a girl to slap her partner if they deserve it”; “It is no big deal if a girl shoves her partner”; “Sometimes girls just cannot stop themselves from punching their partners”; “Sometimes partners deserve to be slapped by their girlfriends”; “Sometimes a girl must hit her partner so that they will respect her”; “Some girls have to hit their partners to make them listen”. Different selections were made for men and women based on the items that were deemed to be most pertinent. Each item’s response categories form a 5-point Likert scale, with 3 corresponding to “neither agree nor disagree” and 5 corresponding to the most permissive attitudes toward IPV in most cases. Reverse coding was conducting in cases where a 5 corresponded to less permissive attitudes. For both male and female participants, an average of all responses was calculated to generate a score ranging from 1 to 5, with 5 representing the most permissive attitudes.

#### 2.2.3. Gender Discrepancy and Gender Discrepancy Stress

While the demographic questions ask about a participant’s sex, the section of the survey on gender discrepancy and discrepancy stress was concerned with socially conditioned gender roles. Gender discrepancy can be understood as the perceived gap between societally determined gender norms and an individual’s internally or externally perceived identity. Stress resulting from this discrepancy has been termed gender discrepancy stress [40]. To assess gender discrepancy and gender discrepancy stress, we used items from the scale developed by Reidy et al. (2014). Again, to reduce the survey burden, we used only select items. The truncated scale contains six questions, with five response categories ranging from “strongly agree” to “strongly disagree”. Three questions pertained to gender discrepancy, and three pertained to discrepancy stress. Participants were asked to provide their feelings about given statements. For men, the gender discrepancy items were: “I am less masculine than the average guy”; “Most women I know would say that I am not as masculine as my friends”. and “Most men would think that I am not very masculine compared to them”. For men, the gender discrepancy stress items were: “I worry that people will judge me because I am not like the typical man”; “I wish I was more ‘manly’”; and “Sometimes I worry about my masculinity”. For women, the items substituted references to femininity for references to masculinity, but they were otherwise equivalent. Item responses ranged from 15, and gender discrepancy scores were generated by averaging the scores for the individual items mapping onto that construct. The same process was used to generate discrepancy stress scores. Consequently, an average score of 3 corresponds to a neutral response, and an average score of 5 corresponds to high discrepancy or discrepancy stress.

#### 2.2.4. AUDIT Scores

Drinking behaviors and problem drinking were measured using the Alcohol Use Disorders Identification Test (AUDIT) [41]. The AUDIT scale contains 10 questions. For 8 of those questions, responses range from 0–4, while 2 of the questions have possible scores of 0, 2, and 4. While there are several different sets of response categories for the AUDIT questions, in each case a 0 indicates abstinence or no drinking related harm, while a 4 represents the maximum harm. For example, item 1 (which asks: “How often do you have six or more drinks on one occasion?”) has responses ranging from “never,” or 0, to “daily or almost daily,” or 4. Scores from individual items are summed to produce an AUDIT score. Scores ranging between 0 and 7 suggest that the respondent is either a non-drinker or engages in low-risk consumption, while responses ranging from 8–14 indicate hazardous consumption. Scores of 15 or above indicate likely AUD.

#### 2.2.5. Intimate Partner Violence

IPV can take several forms, including but not limited to physical, psychological, and sexual violence. In order to assess IPV perpetration and victimization across these three categories, we used an adapted and truncated version of the Composite Abuse Scale, which has items corresponding to each of those constructs (physical, psychological, and sexual IPV) [42]. Overall, the questionnaire asked whether participants had ever experienced a series of actions, with separate but otherwise equivalent items asking whether they had ever perpetrated those actions. Responses took a yes/no format. We included four items indicating physical violence (i.e., “Choked me”), two items indicating sexual violence (i.e., “Forced me or tried to force me to have sex”), and one item indicating psychological abuse (“Harassed me by phone, text, email, or social media”). For our purposes, a “case” of IPV victimization was identified if the participant responded “yes” to any of the 7 items asking about victimization, and IPV perpetration was identified if the participant responded “yes” to any of the 7 items asking about perpetration.

#### 2.2.6. Statistical Analyses

Given the comparative aim of this study, two separate modeling processes were conducted: one for the Ugandan participants and one for the U.S. participants. In each case, analyses began with examination of the univariate distributions of the predictors and the outcome of interest (intimate partner violence perpetration), followed by the examination of the bivariate distributions of variables representing key relationships (e.g., gender and IPV perpetration). Chi-squares and t-tests were calculated to assess between-country differences in distributions of key variables. After the exploratory phase, inferential analyses were conducted using a series of logistic regression models. First, logistic regression models were fit, including a single predictor with IPV perpetration as the outcome. These bivariable models were followed by a multiple logistic regression model for each country, including all predictors that were significant in the bivariable models. Due to its theoretical importance, we included the AUDIT variable in multivariable models regardless of statistical significance. To control for the observed difference in the distribution of age between the two countries, we also included age as a control.

## 3. Results

### 3.1. Sample Characteristics

Among these participants, the education levels differed significantly (χ^2^ = 53.44; *p* < 0.0001), with the largest single group among Ugandans being those with bachelor’s degrees (49.28%) and the largest single group among U.S. respondents being those with secondary or high school degrees (41.33%). This difference may be partially accounted for by the difference in the distribution of age between the two countries, which was also statistically significant (χ^2^ = 8.53; *p* < 0.0001). Among Ugandans, 10% of the sample was under 21, while 44.44% of the U.S. sample was under 21. Self-rated health also differed by country, with 10% of Ugandans rating their health fair or poor and only 5.56% of U.S. respondents doing so (χ^2^ = 10.36; *p* = 0.0348). U.S. respondents also rated their own subjective social status more highly than did the Ugandans, with mean responses of 51.65 in the U.S. and 43.24 in Uganda (χ^2^ = −3.61; *p* = 0.0003). More details on the analytic sample are provided in Table 1.

### 3.2. Exploratory Findings

A significantly greater percentage of U.S. males reported IPV perpetration compared with Ugandan males (58.06% vs. 42.73%; χ^2^ = 4.74; *p* = 0.0294). A greater proportion of males in both the U.S. and Uganda reported IPV victimization than they did perpetration, though again a significantly greater proportion of U.S. males reported victimization than did Ugandan males (67.74% vs. 51.82%; χ^2^ = 5.29; *p* = 0.0215). Though overall a smaller proportion of females reported IPV victimization and perpetration compared with males in each country, a greater proportion of U.S. females reported both perpetration and victimization, though the between-country difference in proportion reporting victimization was non-significant (Perpetration: 40.00% vs. 14.00%; χ^2^ = 17.43; *p* < 0.0001; Victimization: 55.24% vs. 47.00%; χ^2^ = 1.39; *p* = 0.2382). Gender discrepancy and gender discrepancy stress scores were significantly higher among both male and female U.S. respondents compared with male and female Ugandan respondents. IPV attitudes were more permissive among U.S. respondents of both genders when compared with Ugandan respondents of both genders. While female AUDIT scores were not significantly different between countries, U.S. males had significantly higher AUDIT scores than did Ugandan males (8.08 vs. 4.50; t = −3.48; *p* = 0.0006). Notably, U.S. males were the only group among whom the average AUDIT score was in the harmful drinking range. More details from the exploratory analyses can be found in Table 2.

### 3.3. Inferential Models–Uganda

#### 3.3.1. Bivariable Models

Among Ugandan respondents, the odds of IPV perpetration among males were 4.58 times the odds of perpetration among females in bivariable models (95% CI: 2.32, 9.04; *p* < 0.0001). IPV victimization was strongly associated with IPV perpetration odds (OR = 23.57, 95% CI: 8.87, 62.62; *p* < 0.0001). Gender discrepancy, discrepancy stress, and attitudes toward IPV were all associated with odds of IPV perpetration, with more discrepancy, more stress, and more permissive attitudes toward IPV all being associated with higher odds of perpetration. Among these three constructs, the strongest relationship was between IPV attitudes and IPV perpetration. A positive one-unit difference in IPV permissiveness was associated with a multiplicative increase in IPV perpetration odds of 3.34 (95% CI: 2.05, 5.43; *p* < 0.0001). Relative to no or low-risk drinking, hazardous drinking and likely AUD were not associated with odds of IPV perpetration in bivariable analyses among Ugandan respondents (according to SAS-generated *p*-values, although the 95% confidence interval does not contain 1).

#### 3.3.2. Multivariable Model

After controlling for other predictors, the relationship between male gender and IPV was greater, with the odds ratio growing from 4.58 to 6.23 (95% CI: 2.45, 15.86; *p* = 0.0001). IPV victimization was still the strongest predictor of IPV perpetration, with participants who reported IPV victimization having 23.47 times the odds of perpetration compared with those who did not report IPV victimization (95% CI: 7.79, 70.22; *p* < 0.0001). Contrary to our expectations, gender discrepancy and discrepancy stress were no longer significant predictors of IPV perpetration after controlling for other predictors. Attitudes toward IPV were still significant predictors of IPV perpetration among Ugandan respondents, however, with a positive one-unit difference in permissiveness toward IPV corresponding to a multiplicative change of 2.22 in IPV perpetration odds (95% CI: 1.20, 4.10; *p* = 0.0109). More details regarding bivariable and multivariable logistic regression models predicting IPV perpetration among Ugandan respondents can be found in Table 3.

### 3.4. Inferential Models–United States

#### Bivariable Models

Among U.S. respondents, male gender was associated with higher odds of IPV perpetration compared with female gender in bivariable analysis (OR = 2.08; 95% CI: 1.78, 3.67; *p* = 0.0116). In similar fashion to the Uganda models, IPV victimization was a strong predictor of IPV perpetration odds, with those who reported IPV victimization having 27.81 times the perpetration odds of those who did not report victimization (95% CI: 11.59; 66.76). Like the Uganda models, gender discrepancy, discrepancy stress, and IPV attitudes were all positively associated with odds of IPV perpetration, and IPV attitudes were again the strongest predictor. Unlike the models for Uganda, having a score indicating likely AUD was unambiguously associated with significantly greater odds of IPV perpetration relative to no or low-risk drinking (OR = 10.56; 95% CI: 4.12, 27.08; *p* < 0.0001). However, there was once again a difference in the indication of statistical significance between the SAS generated *p*-value and the 95% confidence interval for the odds ratio associated with the intermediate hazardous consumption range, with the *p*-value indicating non-significance but the confidence interval not containing 1.

### 3.5. Multivariable Models

When controlling for other predictors, male gender was no longer significantly associated with odds of IPV perpetration in the U.S. sample, though IPV victimization was still strongly associated with perpetration odds (OR = 27.40; 95% CI: 9.97, 75.32; *p* < 0.0001). Again, gender discrepancy and discrepancy stress were no longer associated with IPV perpetration once controlling for other predictors. In the U.S. sample, IPV attitudes were also non-significant after controlling for other predictors. Scoring in the likely AUD range was still associated with higher odds of IPV perpetration relative to no or low-risk consumption (OR = 7.11, 95% CI: 2.25, 22.54; *p* = 0.0118). More details regarding bivariable and multivariable logistic regression models predicting IPV perpetration among U.S. respondents can be found in Table 3.

## 4. Discussion

In this study we conducted an analysis of the associations between gender discrepancy, gender discrepancy stress, attitudes toward intimate partner violence, and intimate partner violence perpetration, comparing young adults in the U.S. and Uganda. Both the exploratory and inferential findings of this study are surprising in several ways. First, the only subgroup analyzed among whom a majority did not report IPV victimization was Ugandan females, which is in contrast with the well-supported consensus that women in this low-resource setting are at significantly elevated risk of experiencing violence victimization. It is possible that the online survey format and recruitment strategy for the survey contributed to a sampling bias or that female participants in Uganda may have been hesitant to disclose sensitive information. These questions will need to be further examined in future research to elucidate what underlying factors may have contributed to this discrepancy in our study findings relative to previous research.

Another surprising exploratory finding is that across both countries the prevalence of males reporting IPV victimization was higher than that of females, and that the group among whom the largest majority reported IPV victimization was U.S. males (67.74%). While this prevalence is perhaps higher than expected, the high figure could partly be a function of the way an IPV case was defined in this study, wherein a “yes” to any of the 7 questions asked constituted a case for our analyses. There are also several other plausible explanations, including that the U.S. sample is reflective of greater societal acceptance of IPV toward men in that context or that the U.S. respondents were less impacted by social desirability bias.

In general, the U.S. sample was more permissive of IPV, more likely to report IPV perpetration or IPV victimization, and more likely to report problem drinking. In contrast, the Ugandan sample participants rated their self-perceived social status and health status lower than U.S. respondents, but the difference was surprisingly small. The exploratory findings also indicated that IPV attitudes among young adults in the U.S. are more permissive than those of young adults in Uganda, which may complicate the findings of recent research indicating that IPV attitudes among U.S. respondents were generally less supportive of IPV than those of peers in other countries [21].

These findings were contrary to our expectations and underscore the need for further work in this area. It is possible that the sub-section of the U.S. population that is willing to participate in this kind of survey differs systematically from that sub-section in Uganda. For example, those in the U.S. who are attracted to the survey incentive may be lower SES than average, and the technology and web access requirements for participation may systematically bias the Ugandan sample with respect to SES as well, but in the opposite direction. Exploring this hypothesis is complicated by the fact that no objective SES measure was included. Further research is necessary to disentangle and examine these possible explanations.

With respect to alcohol use and assessment of a potential alcohol use disorder, the findings were also somewhat counterintuitive. In our study, the only group for whom the average AUDIT score was in the problem drinking range was U.S. males, and this high prevalence of harmful alcohol consumption may be a contributor to their relatively higher IPV perpetration prevalence as well. Again, it is unclear if this is a bias introduced by the online survey sampling, an effect of other biases, or in some way a function of the fact that the survey was conducted during the COVID-19 pandemic. Recent research in the U.S. indicates that there was a substantial increase in alcohol use during the COVID-19 pandemic [43] and also a significant increase in Alcohol Use Disorder-related mortality rates during the pandemic [44]. In Uganda, on the other hand, there is some indication that alcohol consumption may have been somewhat reduced during the pandemic, at least among women [45]. These differing patterns may partially explain the relatively higher levels of problem drinking in the U.S. sample, as well as the higher levels of IPV perpetration, but this conjecture would need to be confirmed in future studies.

Across bivariable and multivariable models in both countries, the largest association between risk factors examined was between IPV victimization and IPV perpetration. Reporting IPV victimization was associated with a multiplicative increase in odds of IPV perpetration of approximately 23 times and 27 times in Uganda and the U.S., respectively. This strong association across countries could indicate that the overlap between victimization and perpetration is a cross-culturally consistent pattern related to the cyclical nature of IPV. Because of the strong association between IPV victimization and perpetration, we also fit multivariable models omitting IPV victimization to explore whether the other observed associations changed significantly. In these models, gender stress became statistically significant in the Uganda model (*p* = 0.0344), though the odds ratio only increased moderately (from 1.39 to 1.53). In the U.S. model, removing IPV victimization resulted in IPV attitudes becoming statistically significant (*p* = 0.0153), but the point estimate for the odds ratio changed only a minuscule amount (from 1.71 to 1.72). Therefore, the models including IPV victimization were selected as the final multivariable models for both countries.

In the multivariable model for Uganda, attitudes toward IPV were significantly associated with perpetration after controlling for other correlates, underscoring that these attitudes may be an appropriate target for intervention in this population, In contrast, in the U.S. sample, IPV attitudes did not predict IPV perpetration after controlling for other correlates, but those scoring in the likely AUD range on the AUDIT scale had over seven times the odds of IPV perpetration compared with those whose scores indicated no or low risk consumption. This provides further support for the development and testing of IPV interventions that also targe alcohol use among young adults in the U.S.

The findings of this study should be interpreted with some caution for several reasons, as there are important limitations that may have impacted the findings in unknown ways. Since numerous other studies indicate that young women in SSA are at significant risk of a range of negative health outcomes, including IPV victimization, it is possible that the relatively lower prevalence of IPV among Ugandan females in this study compared with other groups is a function of the selection bias regarding the online study participation, a reluctance to report violence due to safety concerns or stigma, or a function of different perceptions of what constitutes violence when compared with male Ugandans or U.S. respondents. Due to the cross-sectional nature of this study, causal attributions cannot be made regarding the associations between the correlates examined and IPV. We have not seen any experimental studies using the Taylor paradigm in Uganda or the broader region, and future longitudinal and experimental research using that approach to examine contextual factors related to neighborhood disadvantage may help to clarify these relationships [46]. Finally, it is possible that the findings of this study are impacted by the nature of the online sampling process conducted, wherein those willing and able to complete online surveys may differ systematically from those who will not or cannot do so. It is unclear what the impact of that potential sampling bias may have been, and it may also have been more significant in Uganda where online surveys are less common than in the U.S.

## 5. Conclusions

While it is generally assumed that individuals are more at risk of violence, harmful drinking, and other negative outcomes in low-income countries, these findings should caution against the assumption that this is universally the case. Overall, however, the findings of this study indicate that IPV perpetration is highly prevalent among young adults in both the U.S. and in Uganda, and that there may be cross-cultural differences in the mechanisms through which IPV occurs. These findings should encourage further longitudinal and experimental research to inform the scientific understanding of cultural differences in the causes of IPV and support the development of more robust, culturally appropriate, and effective interventions to prevent and ameliorate the consequences of IPV in the U.S. and Uganda.

## Figures and Tables

**Table 1 ijerph-19-13373-t001:** Analytic sample characteristics: Sex, age, health, education, and social status among survey participants in Uganda and in the U.S. (n = 408).

Variable	Uganda (n = 210) N (%)	US (n = 198) N (%)	χ^2^	*p*-Value
Sex Female Male	100 (47.62) 110 (52.38)	105 (53.03) 93 (46.97)	1.19	0.2746
Highest level of education * Primary/Middle School Secondary/High School U. Diploma/Associate’s Bachelor’s Degree Master’s Degree	3 (1.44) 46 (22.01) 56 (26.79) 103 (49.28) 1 (0.48)	13 (6.63) 81 (41.33) 46 (23.47) 42 (21.43) 14 (7.14)	53.44	<0.0001
Health Excellent Very Good Good Fair Poor	45 (21.43) 81 (38.57) 63 (30.00) 20 (9.52) 1 (0.48)	64 (32.32) 80 (40.40) 43 (21.72) 11 (5.56) 0 (0)	10.36	0.0348
	Mean (SD)			
Age	23 (1.81)	21.24 (2.34)	8.53	<0.0001
Social Status (0–100)	43.24 (22.62)	51.65 (23.85)	−3.61	0.0003

* In the final analytic data set, 1 Ugandan and 2 Americans responded “other” for their education level. These participants were included in the sample, but their responses for that question were not included in inferential analyses.

**Table 2 ijerph-19-13373-t002:** Distributions of key variables in Uganda and U.S., stratified by sex (n = 408).

Variable	Uganda	United States	χ^2^	*p*-Value
	N (%)			
IPV Perpetration Male Yes No Female Yes No	47 (42.73) 63 (57.27) 14 (14.00) 86 (86.00)	54 (58.06) 39 (41.94) 42 (40.00) 63 (60.00)	4.74 17.43	0.0294 <0.0001
IPV Victimization Male Yes No Female Yes No	57 (51.82) 53 (48.18) 47 (47.00) 53 (53.00)	63 (67.74) 30 (32.26) 58 (55.24) 47 (44.76)	5.29 1.39	0.0215 0.2382
	Mean (SD)		t-value	
Gender Discrepancy Male Female	2.79 (1.28) 2.62 (1.29)	3.37 (1.12) 3.17 (1.08)	−3.40 −3.32	0.0008 0.0011
Discrepancy Stress Male Female	2.55 (1.34) 2.48 (1.29)	3.16 (1.09) 2.90 (1.10)	−3.53 −2.52	0.0005 0.0215
IPV Attitudes Male Female	2.20 (0.63) 1.89 (0.75)	2.53 (0.81) 2.24 (0.77)	−3.23 −3.28	0.0014 0.0012
AUDIT Score Male Female	4.50 (5.49) 3.81 (4.98)	8.08 (8.98) 5.07 (6.55)	−3.48 −1.54	0.0006 0.1251

**Table 3 ijerph-19-13373-t003:** Results of bivariable and multivariable logistic regression models predicting IPV perpetration (n = 408).

	Uganda		USA	
Correlates	OR (95% CI)	Adj. OR (95% CI)	OR (95% CI)	Adj. OR (95% CI)
Gender Male Female	4.58 (2.32, 9.04) Ref	6.23 (2.45, 15.86) Ref	2.08 (1.78, 3.67) Ref	1.35 (0.60, 3.02) Ref
IPV Victimization Yes No	23.57 (8.87, 62.62) Ref	23.47 (7.79, 70.22) Ref	27.81 (11.59, 66.76) Ref	27.40 (9.97, 75.32) Ref
Gender Discrepancy	1.33 (1.05, 1.69)	0.85 (0.53, 1.38)	1.65 (1.25, 2.18)	1.09 (0.63, 1.91)
Discrepancy Stress	1.53 (1.21, 1.93)	1.39 (0.88, 2.19)	1.77 (1.33, 2.35)	1.38 (0.78, 2.42)
IPV Attitudes	3.34 (2.05, 5.43)	2.22 (1.20, 4.10)	2.15 (1.46, 3.16)	1.71 (0.97, 3.00)
AUDIT Score Likely AUD Hazardous Drinking Low risk/No Drinking	4.14 (1.20, 14.30) * 4.51 (2.01, 10.12) * Ref	3.01 (0.34, 26.42) 2.30 (0.76, 6.90) Ref	10.56 (4.12, 27.08) 2.40 (1.09, 5.27) * Ref	7.11 (2.25, 22.54) 2.54 (0.84, 7.69) Ref
Age	0.99 (0.84, 1.16)	0.86 (0.67, 1.12)	0.93 (0.83, 1.05)	0.85 (0.71, 1.02)
Social Status (0–100)	1.01 (0.99, 1.02)		0.99 (0.98, 1.01)	
Education level	0.68 (0.48, 0.96)	1.18 (0.69, 2.02)	0.95 (0.73, 1.24)	
Health Fair/Poor Excellent/Good	2.46 (0.99, 6.14) Ref		0.59 (0.17, 2.08) Ref	

* While the 95% CI does not contain 1, SAS proc logistic generated *p*-values that were >0.05.

## Data Availability

The data presented in this study are available on request from the corresponding author. The data are not publicly available due to privacy concerns.

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
