# Peer review of "A Comparative Analysis of Gender Discrepancy Stress, Attitudes toward Intimate Partner Violence, and Perpetration among Young Adults in the USA and Uganda"

_ijerph, 2022, doi:10.3390/ijerph192013373_

Round 1
Reviewer 1 Report
1. Line 450: "Partner Violence" does not need to be capitalized.
2. Lines 55-64: There is discussion about the SASA! intervention, but there needs to be more details given. From someone who is not familiar with this, I still don't know what it did. What kind of community mobilization efforts? What is the diffusion of innovations theory? More explanation is needed here.
3. Line 67: change "if" to "of"
Measures Section
4. What were the specific questions from the Attitudes Toward Dating Violence Scales? Why are they different based on sex?
5. What were the specific questions for gender discrepancy and gender discrepancy stress?
Discussion
6. Lines 353-358: Need to expand on why men reported more victimization than women. Could it be increased societal acceptance of IPV toward men? Could these be same-sex relationships?
7. Lines 359-366: Same here. Need more discussion as to potential reasons this was found. Why do you think the US would be more permissive of IPV, more likely to report IPV perpetration or victimization, & more likely to report problem drinking compared to the Ugandan sample?
Overall
8. I would consider being more streamlined in your use of "gender" and "sex". In the text, it is referred to as gender (e.g., "male gender" and "female gender") but in the tables it is listed as sex.
Author Response
We would like to thank the reviewer for the helpful comments. We recount the specific revisions made below.
Reviewer 1:
- Line 450: "Partner Violence" does not need to be capitalized.
We thank the reviewer for this correction. The text has been amended accordingly.
- Lines 55-64: There is discussion about the SASA! intervention, but there needs to be more details given. From someone who is not familiar with this, I still don't know what it did. What kind of community mobilization efforts? What is the diffusion of innovations theory? More explanation is needed here.
We have added some language to clarify the nature of the community mobilization entailed within the SASA! study. That study provides important contextual information regarding existing interventions within Uganda but is not the main focus of our paper. Therefore, we kept that section brief. Diffusion of innovations theory is not a critical topic to understand the study conducted in this paper, so the reference to it was removed.
- Line 67: change "if" to "of"
We thank the reviewer for this correction. The text has been amended accordingly.
Measures Section
- What were the specific questions from the Attitudes Toward Dating Violence Scales? Why are they different based on sex? ​
The individual items used are now described in detail, and a brief explanation provided of the difference based on sex.
- What were the specific questions for gender discrepancy and gender discrepancy stress?
The individual items are now described in detail.
Discussion
- Lines 353-358: Need to expand on why men reported more victimization than women. Could it be increased societal acceptance of IPV toward men? Could these be same-sex relationships?
While we feel the need to be careful about providing explanations for the observed phenomena beyond what the data tell us, we have added some discussion of the possible explanations to this section, including differences between the US and Uganda in both social desirability bias and acceptance of IPV. Future work should certainly explore these questions further. Because we do not have information about the sex of partners, we refrained from weighing in on that possibility.
- Lines 359-366: Same here. Need more discussion as to potential reasons this was found. Why do you think the US would be more permissive of IPV, more likely to report IPV perpetration or victimization, & more likely to report problem drinking compared to the Ugandan sample? ​
We have now included a discussion of the possible explanation of these findings.
Overall
- I would consider being more streamlined in your use of "gender" and "sex". In the text, it is referred to as gender (e.g., "male gender" and "female gender") but in the tables it is listed as sex.
The manner in which we refer to gender and sex is quite specific. In each instance where the participants’ demographic identity is under discussion, we use the term “sex,” and include the categories of males and females (or men and women). In each case where we are referring to culturally conditioned roles and experiences based on sex, we use the term gender. Therefore, no changes were made.
Reviewer 2 Report
The topic chosen is of great significance. The style of writing and quality of the research article is good.
However, few points that could have been taken care of are:
1. Theoretical background is lacking. Especially culture specific content will be enormous to report.
2. Sampling technique and research design are elaborately described.
3. Before getting used to the abbreviations two or three places you could have used the expansion ofIPV.
4. In the abstract the last line on general guidelines on keywords need to be deleted.
Author Response
We would like to thank the reviewer for their helpful comments. We recount the revisions below.
- “Theoretical background is lacking. Especially culture specific content will be enormous to report.”
We thank the reviewer for this comment. While one of the key challenges in this study is that fairly little comparative work has been done exploring differences in cultures with regard to gender attitudes and IPV, we have expanded our literature review and discussion of the cultural context related to gender norms, particularly in Uganda.
- “Sampling technique and research design are elaborately described.”
We do not clearly understand the reviewer’s meaning in this comment. It is possible that they are suggesting that our sampling technique and research design are “too” elaborately described. It is also possible that there is a word missing, and that the reviewer intends to say that our sampling technique and research design are “not” elaborately described. We judge the latter possibility to be more likely and have responded to the revision request accordingly.
With regard to our sampling technique, we have amended line 139 to specify that sampling was handled by Qualtrics. Because this is a Qualtrics survey wherein that organization was responsible for participant recruitment, the sampling technique is not possible to further define.
With regard to our research design, we have amended the text to specify that our study is a cross-sectional survey research study.
- “Before getting used to the abbreviations two or three places you could have used the expansion of IPV.”
Our current approach with the abbreviation of IPV is to spell out the phrase once and provide the acronym in parentheses, thereafter referring to IPV using only the acronym. This practice is standard within several prominent style guides, including APA. We could find no style guide that suggests using the expanded version of an acronym “two or three times,” and have therefore made no changes.
- “In the abstract the last line on general guidelines on keywords need to be deleted.”
We thank the reviewer for noticing this oversight. We have deleted the keywords from the abstract.